# Biomimetic Synthesis of PANI/Graphitic Oxidized Carbon Nitride for Supercapacitor Applications

**DOI:** 10.3390/polym14183913

**Published:** 2022-09-19

**Authors:** Manuel Eduardo Martínez-Cartagena, Juan Bernal-Martínez, Arnulfo Banda-Villanueva, Javier Enríquez-Medrano, Víctor D. Lechuga-Islas, Ilse Magaña, Teresa Córdova, Diana Morales-Acosta, José Luis Olivares-Romero, Ramón Díaz-de-León

**Affiliations:** 1Research Center in Applied Chemistry (CIQA), Enrique Reyna Hermosillo, No. 140, Col. San José de los Cerritos, Saltillo 25294, Mexico; 2Laboratory in Biomedicine and Nanotechnology, Cañada Honda, No. 129, Ojo Caliente, Aguascalientes 20196, Mexico; 3Instituto de Ecología, A.C., Red de Estudios Moleculares Avanzados, Clúster Científico y Tecnológico BioMimic®, Campus III, Carretera Antigua a Coatepec, No. 351, Xalapa 91070, Mexico

**Keywords:** polyaniline, graphitic carbon nitride, supercapacitor, conductive polymer, biomimetic synthesis

## Abstract

Polyaniline (PANI) composites have gained momentum as supercapacitive materials due to their high energy density and power density. However, some drawbacks in their performance remain, such as the low stability after hundreds of charge-discharge cycles and limitations in the synthesis scalability. Herein, we report for the first time PANI-Graphitic oxidized carbon nitride composites as potential supercapacitor material. The biomimetic polymerization of aniline assisted by hematin, supported by phosphorous and oxygen-modified carbon nitrides (g-POCN and g-OCN, respectively), achieved up to 89% yield. The obtained PAI/g-POCN and PANI/g-OCN show enhanced electrochemical properties, such as conductivity of up to 0.0375 S/cm, specific capacitances (*C_s_*) of up to 294 F/g (at high current densities, 5 A/g) and a stable operation after 500 charge-discharge cycles (at 3 A/g). In contrast, the biomimetic synthesis of Free PANI, assisted by stabilized hematin in cosolvents, exhibited lower performance properties (65%). Due to their structural differences, the electrochemical properties of Free PANI (conductivity of 0.0045 S/cm and *C_s_* of up to 82 F/g at 5 A/g) were lower than those of nanostructured PANI/g-POCN and g-OCN supports, which provide stability and improve the properties of biomimetically synthesized PANI. This work reveals the biomimetic synthesis of PANI, assisted by hematin supported by modified carbon nitrides, as a promising strategy to produce nanostructured supercapacitors with high performance.

## 1. Introduction

The first enzymatic polymerization was reported in 1951 by Parravano when he achieved the synthesis of polymethylmethacrylate [1]. By contrast, the first enzymatic polymerization of an aniline derivative was reported in 1996 by the group of Romero et al. [2]. The catalytic activity of enzymes is directed towards very specific reactions; for example, horseradish peroxidase (HRP) has been widely used by various research groups for the oxidative polymerization of phenols and anilines. Peroxidase activity is optimal at neutral pH; however, under these polymerization conditions, the synthesis of polyaniline (PANI) presents crosslinking and branching problems due to the preference for ring ortho substitution and consequently ring multi-substitution [3]. On the other hand, multiple polymerization studies show that the substitution is more favorable at pH ˂ 4, but under these conditions, HRP and many other peroxidases denature and lose their catalytic capacity [4]. To overcome this drawback, various groups have evaluated the role of molecular templates in obtaining PANI free of branches or structural defects. Romero et al., reported as well the enzymatic polymerization of template-free PANI employing used soybean coat peroxidase (SBP) as a catalyst [5]. The role of peroxidases is precisely to increase the reaction rate by several orders of magnitude; however, the high costs and difficult handling of enzymes are the main obstacles to the enzymatic method. For these reasons, biomimetic synthesis inspired by enzymatic polymerization is an attractive objective of several research groups [4]. The heme group is common in enzymes with peroxidase activity, such as hemoglobin and myoglobin [6]. In 2000, Gonsalves et al., reported the biomimetic synthesis of polyphenols catalyzed by Hematin (hydroxyferroprotoporphyrin) polyethylene glycol (Hematin-PEG) [7]. Hematin-PEG showed catalytic activity in the pH range from 1 to 4, with an emeraldine-type polymer [8]. Nabid et al., used tetra (*p*-sulfonatophenyl) porphyrin as a biomimetic catalyst, showing optimal catalytic activity in a pH range between 3 and 4 [9,10]. Hu et al. [11] demonstrated the use of hemoglobin as a biomimetic catalyst in the synthesis of polyaniline (PANI)/sulfonated polystyrene (SPS). The spectroscopic characteristics of PANI/SPS were similar to those obtained for enzymatically produced PANI [12]. However, the electrical conductivity decreased slightly at 10^−3^ S/cm. In contrast, Romero et al. [13,14] extended the investigated reaction conditions, pioneering on biomimetic synthesis using catalyst supports. Zhang et al. [15] reported the Hematin-graphene oxide-carbon nanotubes (H-GO-CNTs) composite to construct a dual biosensor to simultaneously detect H_2_O_2_ and tryptophan (Trp), ascorbic acid (AA), dopamine (DA) and uric acid (UA).

Currently, a promising option for energy storage focuses on developing supercapacitors [16,17], an area in which PANI has been extensively used and studied [18,19,20]. When used as supercapacitors, PANI acts as the active material in charge of the storage through REDOX reactions that take place in the structure between different oxidation states. PANI supercapacitors have also achieved high specific capacitance of up to 950 F/g, due to REDOX phenomena involving the polymer’s volume in the charge storage [18].

The electrochemical performance of PANI is highly dependent on its structure and morphology, e.g., PANI agglomerates lead to inefficient polymer performance [21,22]. A way to improve the electrochemical performance of PANI is by incorporating carbonaceous supports, such as carbon nanotubes, graphene oxide, reduced-graphene-oxide (rGO), graphene, activated carbon, porous amorphous carbon, etc. Such composites improve the electrochemical stability of the polymer and favor the storage of the charge due to the associated increase in surface area, which are contributions from the surface chemistry and polymer-support interactions [18].

Porous carbon materials are widely used to increase the stability and conductivity of PANI. They exhibit a large surface area, good chemical stability, and easy processability, which improve the intrinsic disadvantages of the polymer. Furthermore, the introduction of porous carbon materials conveys purely electrostatic electrochemical double layer interactions [23]. In this regard, Yang et al., synthesized a mesoporous PANI/carbon composite, with a high surface area and good cyclic stability reaching up to 517 F/g [24]. Yan et al., prepared supercapacitive electrodes based on graphene nanosheets, carbon nanotubes, and PANI through in-situ synthesis. The electrode showed very high capacitance (1035 F/g, 1 mV/s) and excellent stability (6% loss of capacitance after 1000 cycles) [25]. Some PANI/graphene oxide (GO) nanostructured composites have shown high capacitances close to 555 F/g and high retention capacity (92%, 2000 cycles). On the other hand, various studies of GO-PANI, rGO-PANI and rGO-functionalized/PANI composites show dramatic changes between the polymer-sheet interaction and the electrochemical properties exhibited by the composite, showing that the properties of the supercapacitor, conductivity, and morphology were notably improved for composites composed of superficially modified GO with N-rich groups; leading to composites thermally more stable and with higher specific capacitance than composites formed only by van der Waals interactions [18,26].

Graphitic carbon nitride (typically denoted as g-C_3_N_4_) has become an important material with a wide range of applications commonly focused on photochemical catalysis [27]. However, recently several reports show new synthesis pathways such as pyrolysis (which occurs at a high temperature and requires a controlled atmosphere) and solvothermal synthesis. Herein, PANI/g-C_3_N_4_ composites have been explored in terms of electrochemical properties. Notably, Electrochemical Impedance Spectroscopy (EIS) evaluation shows that the two-dimensional nanomorphology coupled with the chemical characteristics of the g-C_3_N_4_ significantly increases the ionic mobility of the species in the electrode, denoting its high potential as supercapacitors. Considering the few reported works of supercapacitive composites using g-C_3_N_4_ as support, herein we demonstrate the biomimetic synthesis of PANI, assisted by hematin supported on modified g-C_3_N_4_. We examine the electrochemical capabilities of the obtained composites and their potential application as supercapacitors.

## 2. Materials and Methods

### 2.1. Materials

Urea (CH_4_N_2_O, Proquimsa, Coahuila, Mexico), phosphorus pentoxide (P_2_O_5_, Sigma-Aldrich, St. Louis, MO, USA), zinc powder (Chemical Dynamics, Florida, USA), ethanol (C_2_H_6_O, Sigma-Aldrich, St. Louis, MO, USA), methanol (CH_4_O, Sigma-Aldrich, St. Louis, MO, USA), acetone (C_3_H_6_O, Sigma-Aldrich, St. Louis, MO, USA), Toluene (C_7_H_8_, Sigma-Aldrich, St. Louis, MO, USA), Hexane (Sigma-Aldrich, St. Louis, MO, USA), Phosphoric Acid (H_3_PO_4_, Fermont, Nuevo León, Mexico), Ammonium Hydroxide 30% (NH_4_OH, Fermont, Nuevo León, Mexico), Hydrochloric Acid 30% (HCl, Fermont, Nuevo León, Mexico), porcine hematin (Sigma-Aldrich, St. Louis, MO, USA), aniline (Sigma-Aldrich, St. Louis, MO, USA), sodium carbonate (Na_2_CO_3_, Sigma-Aldrich, St. Louis, MO, USA), sodium bicarbonate (NaHCO_3_, Sigma-Aldrich St. Louis, MO, USA), *p*-toluene sulfonic acid (TSA, Sigma-Aldrich, St. Louis, MO, USA), hydrogen peroxide (H_2_O_2_, Sigma-Aldrich, St. Louis, MO, USA), Dimethyl sulfoxide (DMSO, Sigma-Aldrich, St. Louis, MO, USA), *N*-Methyl-2-pyrrolidone (NMP, Sigma-Aldrich, St. Louis, MO, USA), deuterated sulfuric acid (D_2_SO_4_, Sigma-Aldrich, St. Louis, MO, USA), deuterated DMSO (Sigma-Aldrich, St. Louis, MO, USA), deuterated acetone (Sigma-Aldrich, St. Louis, MO, USA) and deuterated water (D_2_O, Sigma-Aldrich, St. Louis, MO, USA).

### 2.2. Methods

#### 2.2.1. Hematin Deposition on Doped g-OCN

10 mg of porcine hematin were dissolved in 20 mL of a buffer solution (bicarbonate/carbonate pH 10, 0.5 M), and magnetically stirred for 30 min in a 40 mL vial. Then 500 mg of g-OCN or g-POCN variant was added, and the vial was immersed in an ultrasonic bath for 15 min. Then, the solution was stirred magnetically at 1000 rpm for 24 h and centrifuged at 14,000 rpm for 30 min. The precipitated dark green solid was filtered (Teflon filter 0.25 µm) and washed 4 times with carbonate buffer solution (pH 10). The obtained solid was dried for 24 h at 40 °C and stored in the dark.

#### 2.2.2. Biomimetic Synthesis Assisted by Hematin Supported in Doped g-OCN

PANI was synthesized using a biomimetic strategy as follows: 100 mg of doped Hematin/g-OCN (or Hematin/g-POCN) were dispersed by ultrasound (40 Hz) in 20 mL of deionized water for 5 min at 25 °C, subsequently, *p*-toluene sulfonic acid is added to the suspension to decrease the pH to 1. Then, the dispersion was stabilized for 2 h under magnetic stirring, and 200 µL of aniline was added to the system at constant stirring (1000 rpm for 12 h). Then, 500 µL of 30% H_2_O_2_ are micro-dosed to the suspension by using a peristaltic pump for 9 min at 0 °C. Once the addition was completed, the temperature and stirring were maintained for 12 h. The recovered dark green solid was filtered (Teflon filter 0.25 µm) and washed with deionized water, ethanol and acetone until the filtrate was colorless. The obtained solid was dried at 70 °C for 24 h and stored in the dark. Figure 1 represents a general scheme of the synthesized material.

#### 2.2.3. Biomimetic Synthesis Assisted by Free Hematin (Cosolvents)

A PANI blank (Free PANI) was prepared by the Biomimetic route. Typically, 10 mg of Hematin are dissolved in 1 mL of DMSO and stirred magnetically at 1000 rpm for 2 h, subsequently, the solution was added to 40 mL of a binary H_2_O/DMSO solution (4:1) and dispersed by ultrasound (40 Hz) for 5 min at 25 °C. Thereafter *p*-toluene sulfonic acid was added until decrease the pH to 1. Then, the dispersion was stabilized for 2 h under magnetic stirring, and 200 µL of aniline was added to the system at constant stirring (1000 rpm for 12 h). Then, 500 µL of 30% H_2_O_2_ were micro-dosed to the suspension using a peristaltic pump for 9 min at 0 °C. Once the addition was completed, the temperature and stirring were maintained for 12 h. The recovered dark green solid was filtered (Teflon filter 0.25 µm) and washed with deionized water, ethanol and acetone until the filtered liquid was colorless, the obtained material was dried at 70 °C for 24 h and stored in the dark.

#### 2.2.4. UV-Vis

The UV spectra were acquired in a Cintra GBC spectrophotometer (Cintra 20, Scientific Equipment, Victoria, Australia), following the change in absorbance, using deionized water as solvent and quarts cells. The 10 mg of g-POCN was dispersed in 1 mL of deionized water and sonicated for 10 min. The spectra were recorded between 300 and 1100 nm.

#### 2.2.5. Powder X-ray Diffraction (PXRD)

The PXRD patterns were acquired on a Brucker D8 Advance Diffractometer (Billerica, MA, USA) with a Cu Kα radiation source (λ = 1.5418 Å). The powdered samples were placed in a standard sample holder; the measurements were made with an interval of 0.02° at a scanning speed of 10°/min from 2θ = 2° to 82°.

#### 2.2.6. Infrared Spectroscopy (FTIR)

Fourier transform infrared spectra were acquired using a Thermo Fischer Scientific FTIR Spectrophotometer (Watertown, MA, USA) in attenuated total reflectance (ATR) mode with a diamond crystal. The sample did not require preparation, the powder was placed on the surface of the glass and the measurement was carried out. The spectra were acquired taking an average of 32 scans with a resolution of 4 cm^−1^ in a range of 400 cm^−1^ to 4000 cm^−1^.

#### 2.2.7. Thermogravimetric Analysis (TGA)

TGA analysis was performed on a TA SDT Q800 device (TA Instruments, New Castle, DE, USA). A platinum basket was used placing at least 1 mg of material, the heating rate was set at 10°/min between 20–800 °C under N_2_ flow at 300 mL/min.

#### 2.2.8. Transmission Electron Microscopy (TEM)

The TEM images were taken on a Titan Fei Termofischer instrument (Waltham, MA, USA) with an acceleration voltage of 200 kV. The samples were prepared by immersing Lazy-carbon grids in a suspension of PANI/g-POCN in NMP-Isopropanol (1:5) and allowing them to dry in a vacuum oven for 2 h at 60 °C.

#### 2.2.9. X-ray Emitted Photoelectron Spectroscopy (XPS)

Measurements were carried out on a Phi5000 Versa Probell, ULVAC-Phi, Inc. (Kanagawa, Japan). With the analyzer perpendicular to the plane of the sample in constant step energy mode. The XPS analysis chamber was operated at 10^−9^ Torr, which contains an X-ray source of Mg-Kα (100 W), subjected to Ultra High Vacuum (UHV, ultra-high vacuum) conditions. This is achieved with the use of turbo-molecular pumps and ionic pumps supported with medium vacuums previously obtained by rotary oil pumps. The samples were dried prior to analysis in a vacuum oven (0 bar) for 24 h at 100 °C, to ensure that they were free of moisture. The samples were grounded in the presence of an electron gun to avoid charging during the measurements. The samples were deposited as a powder on copper tape and analyzed under UHV. To determine the percentages of the present elements, the general XPS spectra were used.

#### 2.2.10. Volumetric Electrical Conductivity Measurement

To measure the volumetric resistivity of the materials, tablets of 12 mm diameter were prepared on which an electrode was painted with silver varnish in a regular way on both sides, later the electrometer tips were placed, and the resistance of materials was measured and recorded according to the equation: VRV = R × A/L. Where VRV is the volume resistivity in Ω × cm, Ω is the resistance in Ohms, A is the area of the electrode and L is the distance between electrodes. The inverse of VR is equal to the volumetric conductivity expressed in S/cm.

#### 2.2.11. Preparation of the Working Electrode (We)

The working electrodes were prepared by dispersing 4 mg of the material in 1 mL of ethanol/water (1:1), using 10 µL of naphion 0.05% wt as a binder. The mixture was dispersed in an ultrasonic bath for 15 min, obtaining a homogeneous and stable suspension. On a 3 mm diameter vitreous carbon electrode, 10 µL of the suspension was deposited (taking extreme care to measure volumes and the correct application on the exposed area of the electrode) completely covering the work area without the existence of interior cavities or external remnants to compare the amount of active material between experiments. Multiple repetitions of the electrode were fabricated for each material to statistically ensure that the measurements were not biased by human error. The coated electrodes with the active material were dried at 60 °C for 60 min. The counter electrode (CE) consisted of a platinum wire and Ag/AgCl was used as a reference electrode (RE). The electrolyte of choice was a 0.5 M H_2_SO_4_ solution in all tests.

#### 2.2.12. Cyclic Voltammetry (CV)

Cyclic voltammetry was performed on the prepared electrodes according to the previous step using a Biologic SP potentiostat as well as using a three-electrode cell, where the working electrode corresponds to a vitreous 3 mm carbon electrode coated with the materials to be evaluated. The counter electrode (CE) consisted of a platinum wire and Ag/AgCl was used as a reference electrode (RE). The electrolyte of choice was a 0.5 M H_2_SO_4_ solution in all tests. Once the electrodes were submerged in the electrolyte, N_2_ was bubbled in for 20 min. The operating conditions consisted of a potential window of −0.2 V to 0.9 V and sweep speeds from 5 mV/s to 100 mV/s in 10 cycles.

#### 2.2.13. Galvanostatic Charge-Discharge

The galvanostatic charge-discharge test consisted of a chronopotentiometry experiment. The cell configuration and the electrolyte correspond to the aforementioned. In this measurement, the current was set at 0.2, 1, 2, 3, 4, and 5 A/g, and the potential window was −0.2 V to 0.9 V. In the case of the stability test, each material was subjected to 500 charge-discharge cycles imposing a current of 3 A/g. The specific capacitance (*C_s_*) reported in this work was estimated from this measurement, according to the equation:(1)$$C_{s}=\frac{t}{m\Delta V}$$

*C_s_* is the specific capacitance and currently used, *t* is the discharge time, *m* is the mass of the pseudo capacitor material and ∆*V* is the potential window used.

#### 2.2.14. Electrochemical Impedance Spectroscopy (EIS)

EIS was performed regarding the previously described cell and electrolyte configuration. Measurements were performed at an alternating voltage amplitude of 10 mV in the frequency range of 100 mHz to 100 kHz against the open circuit potential (Eoc). The electrical equivalent circuit was fitted using ZSimpWin software (ZSimDemo 3.20d, EChem Software, Ann Arbor, MI, USA).

## 3. Results and Discussion

### 3.1. Hematin Supported on g-OCN or g-POCN

Hematin deposition was carried out as described in the experimental section. Hematin/g-POCN and Hematin/g-OCN have a faint green hue different from the g-POCN and g-OCN in their pristine form (which is pearl white), indicating the adsorption of hematin on the surface. The materials were capable to form stable colloidal suspensions over the entire pH scale. UV-vis spectroscopy was used to characterize the colloidal dispersions, Figure 2a shows the spectrum of Hematin/g-POCN. According to Romero et al. [14], the typical spectrum of hematin shows the Soret peak around 380 nm and the Q band close to 600 nm (beta, alpha), both correspond to four orbital border π-π* transitions. In the case of the Hematin/g-POCN aggregate, the typical absorptions of the g-POCN remain without shifts, showing a wide band around 400 nm. An enlargement of the region from 300 to 700 nm shows the presence of the Soret peak of the hematin at 398 nm, which revealed a bathochromic shift. The shoulder centered at 638 nm, which is attributed to the Q band, also underwent a bathochromic shift towards red [28]. These bathochromic shifts are probably due to the interactions of the rings in the g-POCN sheets and the aromatic rings of the hematin.

In the range of 200 to 250 nm, Hematin absorption peaks suffer a displacement toward lower frequencies, ascribed to the effects of π-π interactions and charge transfers between the metal center of hematin and the g-POCN sheet [12,29,30]. These observations evidenced the interaction between hydroxyferroprotoporphyrin with the oxidized carbon nitride sheets, generating a stable layer to catalyze the oxidative synthesis of PANI.

Figure 2b shows the Hematin/g-POCN diffractogram. As shown, the planes (100) and (002) of g-POCN appear in the conventional positions; the surface elemental concentration was determined by XPS (Figure 2d). Interestingly, the intense plane at 9.2° suggests the presence of hematin molecules on the surface since this same plane can be observed for free hematin at 8.7°. This displacement can be attributed to changes in the hematin molecules ordering over the g-POCN sheets [15,31]. A diffraction maximum at 22.4° may be associated with the same diffraction observed in the diffractogram of free hematin. TEM images of the Hematin/g-POCN composite revealed a bidimensional morphology (Figure 2c), which evidenced the assembly of the composite under the described experimental conditions.

### 3.2. Free PANI and PANI/g-OCN PANI/g-POCN

#### 3.2.1. FTIR

Figure 3a shows the FTIR spectra of PANI and PANI/g-POCN. The characteristic bands of PANI are observed near 3000 cm^−1^, associated with the CH stretch of the aromatic ring in the polymer backbone. The signal at 3250 cm^−1^ also shows the NH stretch, which moved to lower wavenumbers due to intermolecular interactions (hydrogen bonds between polymer chains) and the doping ion effect [32].

The FTIR spectra of PANI/g-POCN shows a 1565 cm^−1^ signal attributed to C=C stretching in the quinoid ring of PANI, and the signal slightly close to 1590 cm^−1^ is attributed to the N-H bending [32]. The signals at 1482 and 1442 cm^−1^ are attributed to the C=C stretch in the benzenoid ring [32]. The signal at 1375 cm^−1^ corresponds to the CN stretch in quinoid-benzenoid-quinoid (QBQ) units [32,33]. the signal at 1332 cm^−1^ is ascribed to the CN stretch, as well as the band at 1285 cm^−1^ which corresponds to the CN stretching of secondary aromatic amine [32]. The signal at 1240 cm^−1^ is associated with CN stretching of benzenoid-benzenoid-benzenoid (BBB) units [32]; meanwhile, the signal of 1103 cm^−1^ corresponds to the stretching of BNB units in the polymer chain [32]. The signal of 1029 cm^−1^ is associated with stretching S=O resulting from doping with TSA in the sample [14]. The bands at 781 and 875 cm^−1^ denote deformation outside the CH plane of 1,2,4 trisubstituted and 1,2 (ortho)monosubstituted rings, respectively [32]. The signals close to 820 cm^−1^ are characteristic of *p*-ring substitutions and the signal at 673 cm^−1^ could be assigned to out-of-plane deformations of the monosubstituted ring [32]. The spectra obtained are in accordance with previous reports, thus rendering the successful formation of PANI on carbon nitride sheets.

#### 3.2.2. Thermal Analysis

The thermogravimetric analysis of Free PANI, PANI/g-OCN, and PANI/g-POCN is shown in Figure 3b. The data reveals that the thermal stability of the three polymers is similar up to 100 °C, where a weight loss occurs attributed to interstitial water molecules. PANI (Figure 2b, red line) initiates thermal degradation at 275 °C [33]. The next degradation step at 275 °C can be attributed to thermolabile groups of the BQ rings. The gradual weight loss after 350 °C corresponds to the degradation of the polymer chains until 600 °C. The total weight loss of 43% is consistent with reported data [34,35].

The PANI/g-OCN and PANI/g-POCN composites (Figure 3b, blue and black lines, respectively) showed a first degradation step of 3.9% due to the water adsorbed on the polymeric matrix. The second weight loss begins at ca. 275 °C, where the typical gradual weight losses occur due to the decomposition of OH, NH, and thermolabile groups of nitride sheets. The third degradation step, near 400 °C, represents the loss of NO groups and probably COOH, PO. The fourth degradation is attributed to the decomposition of the polymer and the destruction of the lamellar structures of carbon nitride oxidized support. The subsequent weight loss corresponds to the massive degradation of the nitrogenous structures, with a final weight loss, at 600 °C, of 25% for PANI/g-POCN and 35% for PANI/g-OCN. Thus, the PANI/g-POCN showed the highest thermal stability, which corresponds well to other PANI/GO composites [36,37,38].

#### 3.2.3. Microstructure Analysis by PXRD, SEM, and TEM

The X-ray diffraction plots of PANI and PANI/g-POCN are shown in Figure 4. The diffraction maxima at 2θ = 27.2° of PANI/g-POCN (black line) corresponds to the plane (002) of graphitic carbon nitrides, indicating the interlaminar stacking distance. It is interesting to note that such reflection shifted to the left with respect to the diffraction peak in the original g-POCN [39], which suggests an interlaminar distance increase due to the intercalation of PANI chains between the g-POCN clusters. The diffractions at 2θ = 15, 20, and 26°, correspond to the planes (011), (020), and (200) of Free PANI, such reflections are characteristic of the pseudo-rhombic cell associated with the emeraldine salt structure [8,33,40], which is also observed in the case of the reference Free PANI (Figure 4, red line).

SEM micrographs obtained for PANI/g-OCN, PANI/g-POCN, and Free PANI are shown in Figure 5. The PANI/g-POCN micrograph (Figure 5b) revealed the formation of microparticles with a size smaller than 5 microns along with large polymer-coated platelets. These features might provide a synergistic effect between the particles with homogeneous distribution and support to increase the surface area in the composite, which could provide better energy storage capabilities for electrochemical capacitors [18]. It is also expected that the electronic molecular interaction associated with the π-stacking, electrostatic interactions, and hydrogen bridges between PANI and g-OCN could significantly affect the composites quality.

On the other hand, the SEM micrograph of Free PANI (Figure 5c) shows polymer clusters around 10 µm, which decreases the stability and availability of the active area. It is not surprising to find these agglomerates in conventional oxidative polymerizations [41], but it is interesting to observe this morphology in the biomimetic synthesis, since it provides additional evidence that the biomimetic synthesis mechanism runs very closely to the conventional oxidative mechanism; a fact that has been widely theorized by some authors [1,41,42,43,44,45]. Thus, we could assume that the factors that influence the oxidative synthesis of PANIs are probably also present in the biomimetic method. 

TEM micrographs of PANI/g-POCN composite (Figure 6) show the presence of micrometer-sized sheets. The growth of PANI on the sheets is heterogeneous, although it completely covers the sheets occasionally. Globules can also be observed as spheroidal particles of the dispersed polymer. This behavior is explained by the differentiated deposit of Hematin on the two-dimensional networks which saturates the surfaces in certain areas and deforms the sheet due to interfacial phenomena. Supramolecular interactions prevent the deposition of Hematin. The composite sheets reach micrometric sizes, which is undesirable for volumetric materials, and in the case of platelet materials, they are expected to exhibit superior properties with respect to their non-nanostructured analog (PANI).

### 3.3. Electrochemical Characterization

The electrochemical performance of the Free PANI, PANI/g-POCN, and PANI/g-OCN electrodes was evaluated by cyclic voltammetry, galvanostatic charge-discharge, and electrochemical impedance spectroscopy.

#### 3.3.1. Cyclic Voltammetry

The CV study was carried out to elucidate the electrochemical behavior of PANI/g-OCN composites and compare it with the one of Free PANI. More specifically, cyclic voltammetry developed in a three-electrode configuration is useful, especially for the study of the pseudocapacitive contributions inherent in the electrode material; this is the most appropriate technique for the study of any contribution of faradic origin present in the system [18]. Figure 7 shows the voltammetric profiles obtained for Free PANI, PANI/g-OCN, and PANI/g-POCN at 100 mV/s with a working window of −0.2 V to 0.9 V in 0.5 M H_2_SO_4_. The voltametric profile of PANI/g-POCN shows quasi-rectangular shapes attributed to the pseudocapacitive behavior of materials. In contrast, Free PANI and PANI/g-OCN voltammogram profiles resemble nearly quasi-reversible systems.

An ideal Electrochemical Double-layer Capacitor (EDLC) generates cyclic voltammograms with a perfect rectangular shape; the presence of faradic reactions leads to the existence of capacitive peaks that generate the distortion of the rectangular shape towards a quasi-rectangular or completely irregular shape [46,47]. Principally, it is expected that the developed carbonaceous support (g-OCNs) allows for an increase in the available surface area capacitance with respect to Free PANI. It is also conducive to expect that the chemical stability of PANI in the composite improves due to the interactions that are generated with the substrate or support based on g-OCNs.

The discrepancy of the 90° angle with respect to the abscissa is governed by the intrinsic resistance of the system. From the results in Figure 7, we could deduce that the least resistive composite is PANI/g-POCN, behavior related to the conductivity promoted by the structural order obtained through g-POCN nanosheets and the oriented growth of the PANI on it. The chemical qualities of the structure’s phosphorous doping consolidate this behavior [48,49]. The voltammograms in Figure 7 show two REDOX oxidation-reduction pairs in the three materials, which correspond to the redox transitions between the leuko-emeraldine form (r-semiconductor)/polaron emeraldine (conductor) and emeraldine base/pernigraniline [50]. The first pair of peaks (oxidation A/reduction A’) is the redox transition from leuko-emeraldine to emeraldine salt, and the second pair of peaks (oxidation C/reduction C’) is associated with the faradaic transition from emeraldine base to pernigraniline [50]. These redox transitions are depicted in Figure 8.

Free PANI presents the most resistive voltammetric profile of the three materials which is verified in the EIS analysis and is largely conditioned by morphology, being the average distance of the electrolyte path during charge-discharge increases. The increase in the particle size distribution of the nanostructured morphologies causes a decrease in the surface area, leading to a smaller available electroactive surface and an increase in the resistance to charge transfer, blocking or slowing down ionic migration [51]. Regarding the change in the area observed between the three voltammograms, it should be noted that they are determined by the capacity of the device. One way to approximate the capacitance of the electrode requires the calculation of the amount of voltammetric charge (QCV) by integrating each voltammogram [52]. The capacitance of the system is equal to the quotient of QCV divided by the mass product of the active material and the working voltage [46]. The composite with the largest area or QCV is the PANI/g-POCN, which presents a capacitance of 387.9 F/g at 10 mV/s; the PANI/g-OCN composite has a profile similar to the PANI/g-POCN, but a lower QCV for the same amount of material analyzed, so its capacitance (342 F/g at 10 mV/s) demonstrated lower velocity of charge propagation through the surface [51]. This is closely linked to the morphological changes of the particles that detract from the available electroactive area. Also, the characteristics of the g-OCN support (higher density of oxidized groups) can influence the mobility of the ionic species in the system. The speed at which the doping-dedoping of the PANI backbone takes place, and the blocking or variation of ionic migration, influence the number of effective charges that are stored throughout the nanodomains (total capacitance). Free PANI exhibits the smallest area of the three Voltammograms, therefore, the smallest capacitance is only 227 F/g at 10 mV/s. We can attribute the latter to a disordered micromorphology. In support of this deduction, several reports have shown that polymerizations that proceed by the conventional oxidative method or by an assisted method (enzymatic, photopolymerization) give rise to globules or microfibers with wide distributions and agglomerations that do not confer an ideal nanometric order [41]. This depends on the processes of charge transfer and ionic mobility (the EIS analysis for Free PANI showed negative results). It should be mentioned that g-OCN polymers (Appendix A) do not contribute significantly to EDLC since they have a very small QCV, unlike other carbon-based materials, such as GO or carbon nanotubes, that increase capacitance by direct contribution to the electrochemical double layer. However, g-OCNs provide an ideal substrate for nano-structuring PANI, serving as a support for the hematin and as a template for the growth of the polymer. g-OCNs substrates increase the surface area because they are thin sheets, and in turn, they give chemical resistance to the composite. In general terms, this is translated as an increase in the specific capacitance observed above; due to the increase in the specific electroactive area that the flat or two-dimensional structures of the g-OCN induce.

#### 3.3.2. Galvanostatic Charge-Discharge

Galvanostatic charge-discharge is the most used technique to determine the storage characteristics of devices; the galvanostatic charge-discharge curve consists of the electrochemical method known as current inversion chronopotentiometry and when applying a loop, it is known as cyclical chronopotentiometry. The technique is based on applying an excitation signal in the form of a constant current in the cell. The response signal is the potential of the working electrode; the measurement of the potential difference with respect to time gives rise to the chronopotentiometric curve [53]. In this work, the measurements were performed in a potential window of −0.2 to 0.9 V vs. Ag/AgCl (0.5 M H_2_SO_4_) setting the current at different values (5 µA–125 µA). Figure 9 shows the galvanostatic charge-discharge curves obtained for the Free PANI and the PANI/g-OCN and PANI/g-POCN composites.

In the three chronopotentiograms of Figure 9d, the load branch has different initial potentials for each system. PANI/g-POCN showed the lowest potential, followed by a slightly higher potential for PANI/g-OCN, and finally Free PANI with an appreciable high potential (associated with the increase in ESR) [54]. However, since it is the same current density applied in each case, the reason is kinetic in nature, which involves a longer onset time for the faradic transitions involved at anodic potentials which indicate behavior dependent on the ionic flow [54].

Applying a constant current to the electrode causes the electroactive species to be oxidized/reduced at a constant rate, so the electrode potential varies according to the time in which the concentration of reactants to products modifies the surface of the electrode [51,55]. Once the processes of charge transfer and mass transfer through the electrode have started, each material takes different times until the electroactive surface has accepted all the charge that it can store at a constant current implying a monotonic increase in potential. Until more electrons can be forced in, the electrode potential changes abruptly from values in the anodic direction to values in the cathodic direction, giving rise to the discharge branch. The overall shape of the curve is governed by the reversibility of the reactions at the electrode [56]. Regarding the discharge branch in the three materials, at the beginning, an ohmic drop is observed due to electronic contribution and ionic contribution [51,57] that allow estimating the system ESR. Out of the three polymers, Free PANI showed the highest ESR associated with this phenomenon, which confirms the observations of CV and EIS where the same had been deduced. This seems to indicate that the polymerization assisted by free hematin does not contribute significantly to improving the intrinsic conductivity of the material. On the other hand, the PANI/g-OCN and PANI/g-POCN composites do not present notable differences in discharge time (Figure 9b,c), which is associated with the increased conductivity of the polymers (see Table 1) and their adequate nanostructure (as observed in XRD and microscopy). The two-dimensional substrate complemented this by increasing electron transfer and ionic diffusion, therefore, improving the performance of the electrolyte-electrode interface.

The increase in discharge time is directly proportional to the increase in the capacitance of the material, asserting that the increase in discharge time involves processes of charge transfer and ionic migration that occur at a higher speed in the case of the composites. Different reports have attributed the phenomenon observed for PANI/g-POCN [36,51,58] to the increase of the surface area and decrease of the material’s intrinsic resistance, ordered nanomorphology, and homogeneous chemical structure along the electroactive polymer chains [23,52,58]. On the other hand, PANI seems to have the shortest discharge time, which is not surprising after characterizing it by CV and EIS, where a high resistance to charge transfer (lower electrical conductivity) and diffusional migration of ionic species was slowed down or hampered by undesirable nanomorphology.

The discharge branch time is universally accepted for the calculation of the material’s specific capacitance. Table 1 shows the specific capacitances (*C_s_*) obtained for the three materials at different current densities. In general terms, the discharge time increases with the decrease in current. At lower currents, the ions are diffused within the electrode, resulting in greater available surface and greater *Cs*. In contrast, with higher current density the *C_s_* are reduced due to the ionic diffusion-limited to the interior of the electrode and governed by the available surface area in the immediate interface to the electrolyte [54]. The highest *C_s_* are shown by PANI/g-POCN. This agrees with the observations of CV, which is linked to the increase in surface area (promoted by the g-POCN sheets), and the improved structural characteristics through growth on g-POCN sheets. It should be noted that the g-POCN material also exhibits satisfactory electronic transfer and mass transfer properties according to the EIS study (Appendix A). Gao et al. [59], pointed out that the presence of two-dimensional materials (such as GO, RGO, and Graphene), in this case, g-OCN and g-POCN, increases the porosity and ionic diffusion through the PANI network and the support. Therefore, the electrical conductivity of the PANI/g-POCN composite increases in comparison to the non-nanostructure materials (Table 1). On the other hand, Free PANI exhibits a lower capacitance, as well as the lowest electrical conductivity of the three materials. This data corroborates the observations made throughout the different electrochemical tests in this study.

Figure 10a shows the comparative graph between the *C_s_* of the three materials, studied at different constant current densities (from 0.2 A/g to 5 A/g). In all cases, the highest *C_s_* were obtained for PANI/g-POCN, which as discussed is promoted by an increase in the speed of electrical and diffusion transfer phenomena due to the nanomorphological and structural qualities of the material. Such phenomena were also observed for PANI/g-OCN but with a substantial decrease in *C_s_* as current density increases (a rate capability of 46%). In contrast, PANI showed an abrupt *C_s_* decline (a low-rate capability of 25%). Comparatively, the PANI/g-POCN maintains high *C_s_* values even at high current densities (25-fold higher current density than the initial value), which indicates that the ionic diffusion rate is not significantly influenced by increasing current density (a good rate capability of 64%). A fact that is explained by the short ionic migration routes and large available surface area that allow to a substantially increase in the electrolyte-electrode interface [59]. In contrast, PANI/g-OCN and Free PANI undergo drastic changes in diffusion velocity with the increasing current, which reduces their storage capacity loading. However, the obtained results for the three materials are high at current densities greater than 3 A/g.

Figure 10b shows the stability graph, expressed as the Capacity Retention, of PANI/g-OCN, PANI/g-POCN, and free PANI during 500 galvanostatic charge-discharge cycles with a current density of 3 A/g. This test shows the retention stability of the specific capacitance during the charge and discharge cycles. As observed, PANI (red line) lost about 50% capacity retention after 500 charge-discharge cycles, which is not suitable for the development of supercapacitors. This behavior is related to crosslinking phenomena, swelling, and degradation by over-oxidation, inherent to the redox reactions elicited through the polymeric network, that involve repeated expulsion and uptake of ions [18]. The PANI/g-OCN (blue line) and PANI/g-POCN (black line) composites revealed the highest retention, losing only ca. 23% of the retention capacity. This could be attributed to the documented stability of other PANI composites [59].

Interestingly, both supports g-OCN and g-POCN showed similar capacity retention, which could be attributed to the nanostructured PANI intensely interacting with the surface of the g-OCN and g-POCN sheets. These phenomena generated composites that preserve a two-dimensional structure provide chemical resistance and structural stability to the PANI network in very low pH environments. Altogether, carbon nitride supports provided an increase in the surface area and *C_s_* and sufficient stability for PANI composites. Thus, the experimental data demonstrated that PANI/g-OCN and PANI/g-POCN composites globally meet the requirements or desirable electromechanical characteristics for materials with potential for application as electrodes in devices of high capacitance electrochemical capacitors, known as supercapacitors.

#### 3.3.3. Electrochemical Impedance Spectroscopy (EIS)

EIS is a highly sensitive technique to measure the electrical response of a chemical system by applying a variable frequency and using alternating current through an electrochemical cell. The main advantage of this technique is that it allows to distinguish and separate all the processes that take place in the electrochemical system [60].

Appendix A and Figure 10d show the Nyquist curves obtained for the Free PANI, and PANI/g-OCN and PANI/g-POCN composites, respectively. The generated impedance spectrum for the three materials has the conventional shape for the capacitive system (high frequencies, mid frequencies, and low frequencies); however, the frequencies of the patterns formed are substantially different. For example, in the case of the PANI/g-POCN and PANI/g-OCN composites, a semicircle appears at high frequencies, which satisfactorily reproduces the behavior of a simple pseudocapacitive electrochemical system, proposed by Brett and Brett in 1993 [61]. The increase in the resistance of the interface with the electrolytic solution and electrode (RΩ) shifts the origin of the curves in the Z’ axis; the intersection of the curve with the X axis at high frequencies represents the internal resistance of the system or the equivalent series resistance (ESR), the sum of the electrolyte resistance, the intrinsic resistance of the active material and the resistance of the interface between the active material and collector. These resistors predetermine the loading and unloading speed [62].

A good approximation of the ESR can be obtained by reading the initial value of Z’ from the Nyquist diagram which is 8.18 Ω for PANI/ g-OCN, 23.6 Ω PANI/g-POCN, and 45.2 Ω for Free PANI (Appendix A). The PANI/g-OCN composite has lower resistance due to the increased intrinsic conductivity of the material.

The PANI/g-POCN composite has two well-defined parts of the curve, a semicircular portion at high frequencies and a straight-line portion that begins at the end of the semicircle and extends to low frequencies, known as the Warburg’s tail [63]. This behavior fits the simplest equivalent circuit, consisting of Rct and Zw in series and parallel to a Cdl, all attached to a resistor Rs (Figure 8d).

The small semicircle at high frequencies is associated with the electronic transfer speed between the electrode and the electrolyte. The resistance to charge transfer can be obtained from the portion of the Z’ axis, the Rct of the PANI/g-POCN is the lowest of the three materials (20.2 Ω). It is followed by the PANI/g-OCN (Rct > 200 Ω), while the Free PANI has the highest resistance (Rct > 5.5 kΩ). According to similar reports for PANI/GO and PANI/RGO composites [59], the low Rct of PANI/g-POCN may be associated with higher conductivity, higher surface area, and the features associated with a two-dimensional morphology. PANI/g-OCN and Free PANI notably increase their resistance to charge transfer, which is closely related to their morphology. In the case of PANI/g-OCN, g-OCN sheets are thicker which reduces the surface area. It also has a high content of O. In the case of Free PANI, the typical globular formations, were observed in this type of polymerization, detract from the electroactive area and the percolation of charges yield materials, low coefficient diffusion, and consequently, high resistance [18,41,54,59]. The straight line that extends with a certain inclination at the end of the semicircle is related to the contribution of the Warburg impedance. In this sense, it is known that in real electrochemical capacitors, a line with a 45° angle extends just after the local minimum of the semicircle at medium frequencies. This is the Impedance involved in the transfer of ionic current to an electrode by a Faradaic process and a measure of the ratio ion diffusion; the one that varies inversely with the square root of the frequency [60]. The Nyquist diagram of g-POCN (Appendix A), presents a finite straight line that extends to high values of Z’, with an angle close to 90°, indicating high ionic mobility through the exposed area. From these results, it can be inferred that the diffusion properties of the ions in the PANI/g-POCN electrode are strongly influenced by the characteristics of the two-dimensional polymer [40,64,65]. Additionally, Appendix A shown relevant studies recently reported and their comparison with the present work.

## 4. Conclusions

By integrating the biomimetic polymerization of aniline, assisted by hematin supported on modified carbon nitrides, we report a strategy to produce PANI/g-OCN and PANI/g-POCN composites with enhanced electrochemical properties and their potential application as supercapacitors. First, we demonstrated the modification of g-C_3_N_4_ substrates with oxygen and phosphorus. Then, as demonstrated by UV-vis and XPS spectroscopies, the modified g-C_3_N_4_ sheets were used to effectively perform hematin deposition and create stable composites able to catalyze the oxidative polymerization of aniline.

The resulting supports exhibited attractive oxidative performance to produce PANI/g-OCN and PANI/g-POCN composites with up to 89% yield. The microstructural analysis of the obtained composites revealed nanostructured materials that benefit the electrochemical capabilities of the materials. In comparison with the non-nanostructured PANI (Free PANI, obtained by biomimetic synthesis assisted by stabilized hematin in cosolvents), we found that the phosphorous- and oxygen-modified CN supports provide superior conductivity features to the composites (0.0171 and 0.0375 S/cm for PANI/g-OCN and PANI/g-POCN, respectively) and stable operation under high current density conditions (*C_s_* up to 294 F/g at 5 A/g for PANI/g-POCN) and after 500 charge and discharge cycles. This work evidenced that nanostructured PANI/g-OCN and PANI/g-POCN composites increase the surface, chemical resistance, and structural stability of the materials, which provides design principles to produce suitable electrochemical PANI composites for the development of high-performance supercapacitors. 

## Figures and Tables

**Figure 1 polymers-14-03913-f001:**
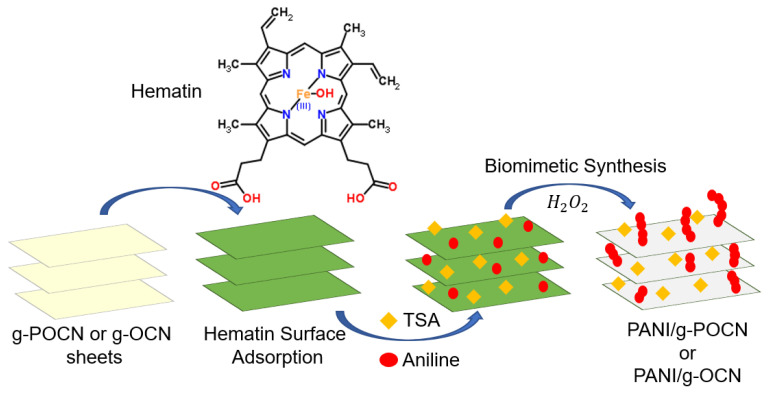
General diagram of the synthesized PANI/g-C3N4 supercapacitor composites.

**Figure 2 polymers-14-03913-f002:**
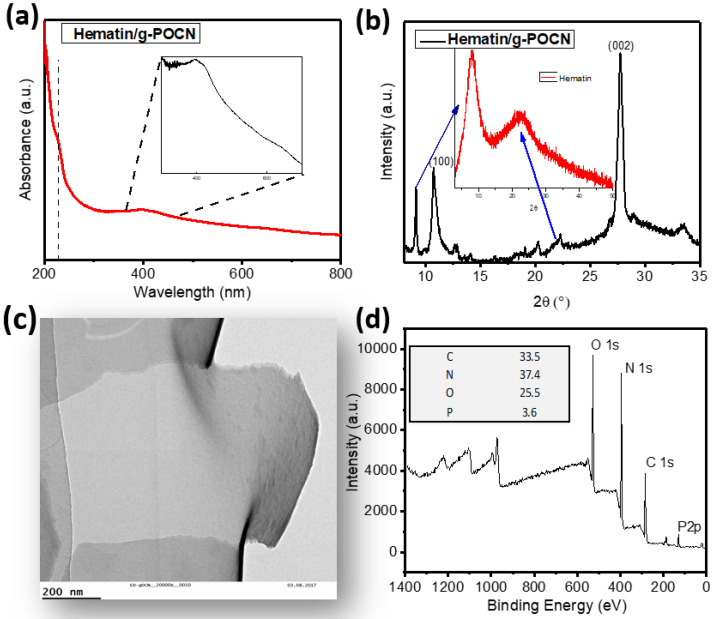
(**a**) UV-Vis spectrum of Hematin Supported in g-POCN, inset: magnification of the 400–700 nm region of the hematin contribution. (**b**) Hematin/g-POCN diffractogram, inset: Hematin diffractogram, diffraction maxima related to hematin. (**c**) g-POCN Transmission electronic micrograph, the detail shows a sheet fragment on an edge. (**d**) g-POCN XPS spectra, inset: percentual surface elemental concentration.

**Figure 3 polymers-14-03913-f003:**
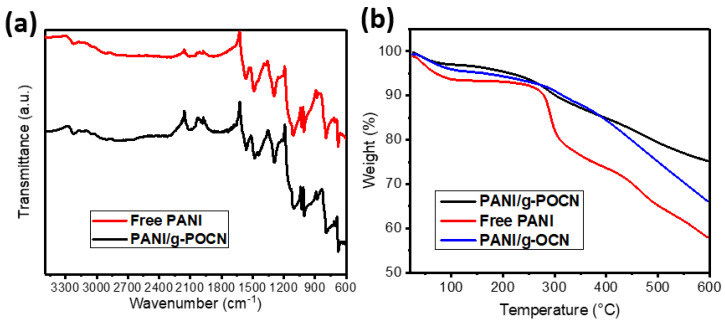
(**a**) Free PANI and PANI/g-POCN composites FTIR spectra. (**b**) Free PANI, PANI/g-OCN and PANI/g-POCN thermograms.

**Figure 4 polymers-14-03913-f004:**
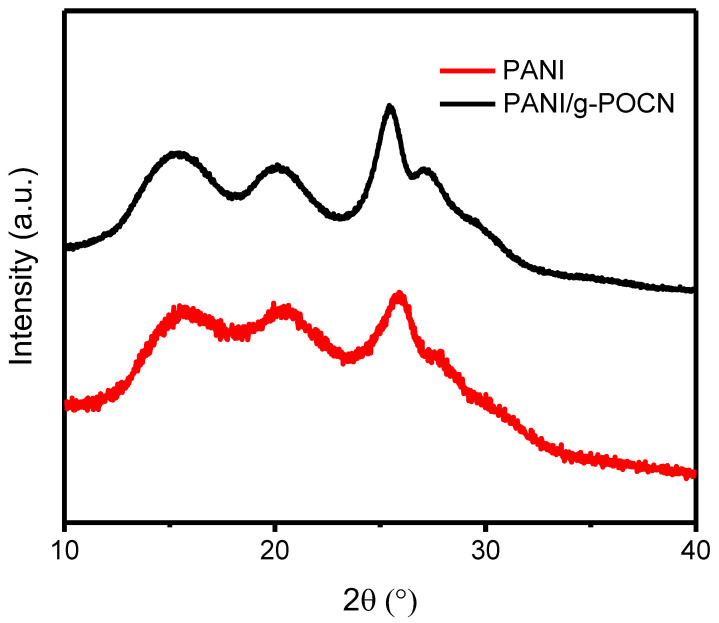
Free PANI and PANI/g-POCN diffractograms.

**Figure 5 polymers-14-03913-f005:**
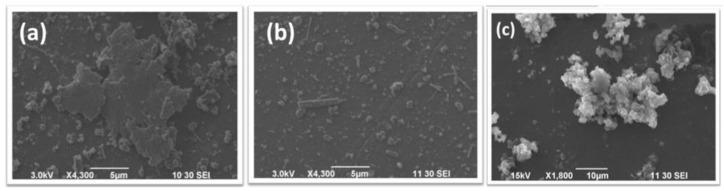
SEM micrographs (**a**) PANI/g-OCN, (**b**) PANI/g-POCN and (**c**) Free PANI.

**Figure 6 polymers-14-03913-f006:**
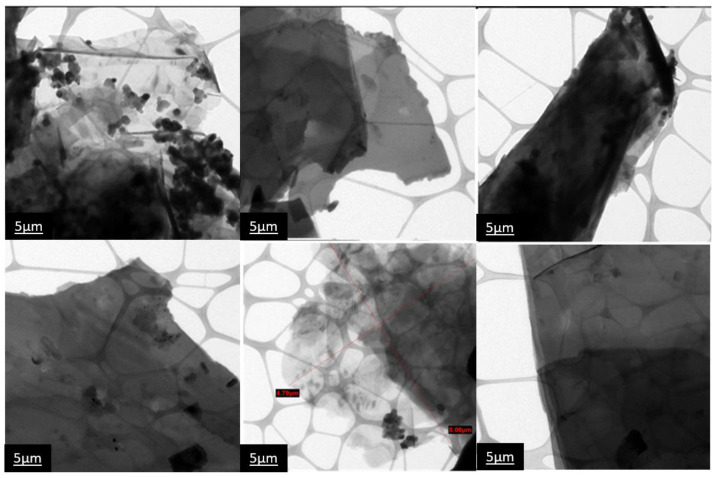
TEM PANI/g-POCN micrographs.

**Figure 7 polymers-14-03913-f007:**
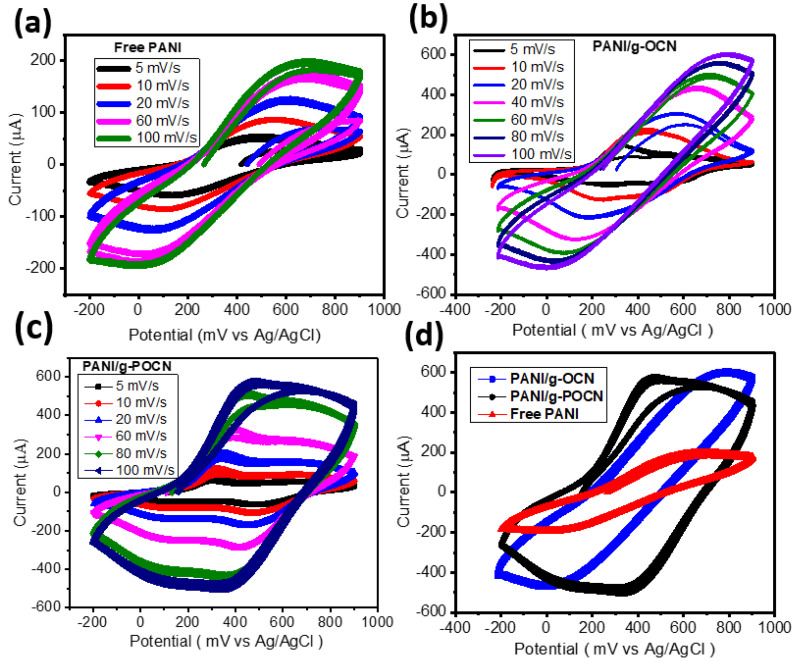
Cyclic voltammetry of the electrodes: (**a**) Free PANI, (**b**) PANI/g-POCN, (**c**) PANI/g-OCN, and (**d**) a comparative plot between the composites.

**Figure 8 polymers-14-03913-f008:**
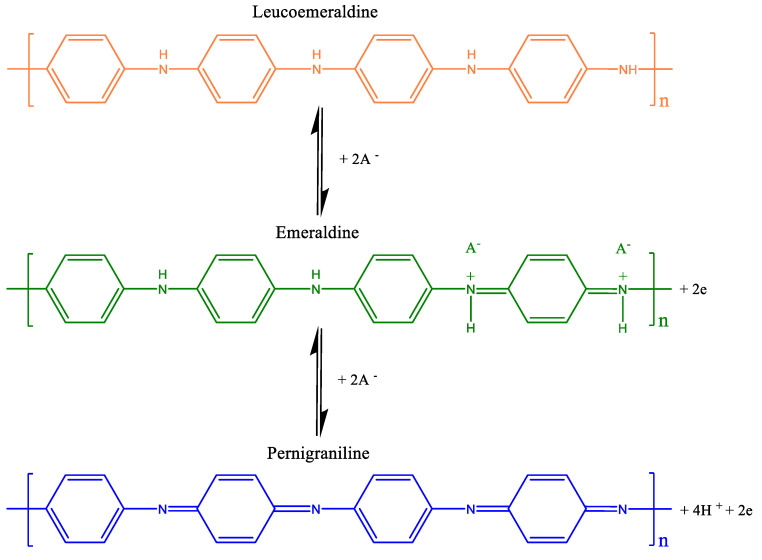
Redox transitions of the polyaniline.

**Figure 9 polymers-14-03913-f009:**
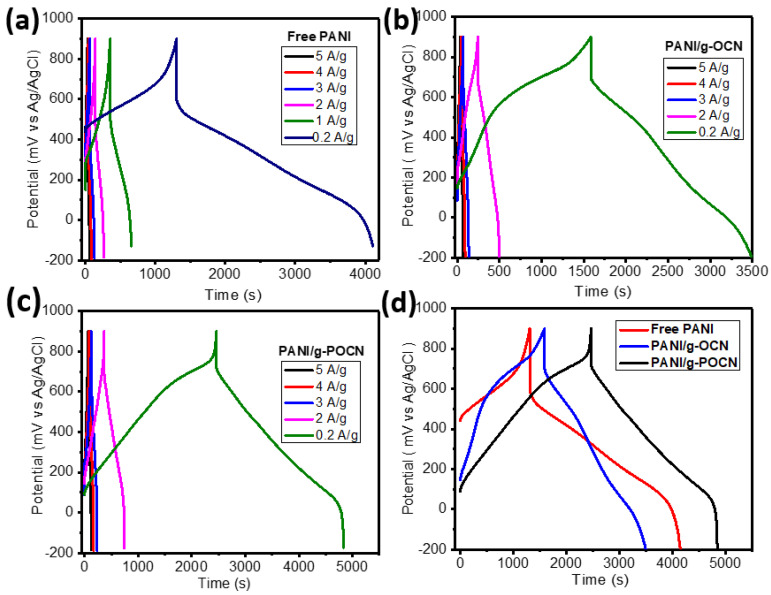
Galvanostatic charge-discharge curves: (**a**) Free PANI, (**b**) PANI/g-OCN, (**c**) PANI/g-POCN, and (**d**) comparison between composites at 0.2 A/g.

**Figure 10 polymers-14-03913-f010:**
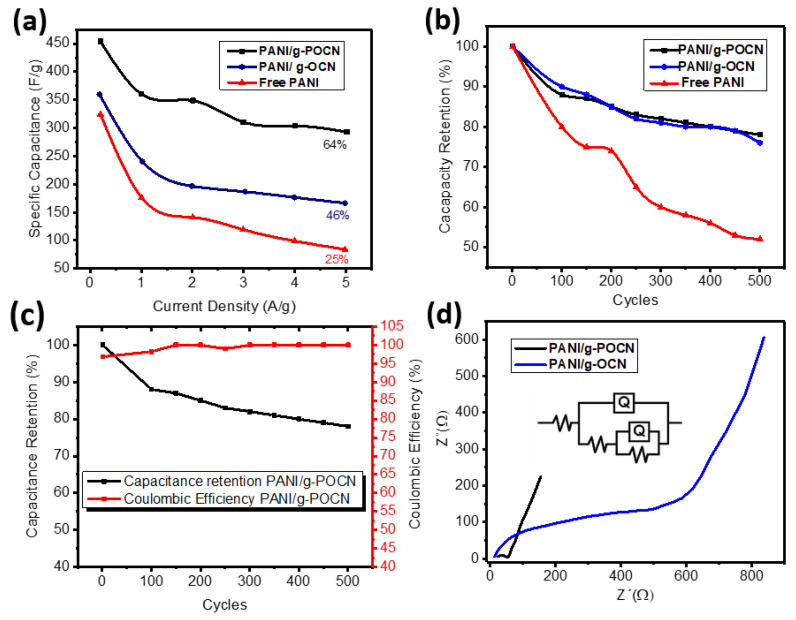
(**a**) Plot of *C_s_* vs. A/g for different currents applied to the materials PANI/g-OCN, PANI/g-POCN, and Free PANI. (**b**) Stability of materials after 500 galvanostatic charge-discharge cycles at 3 A/g. (**c**) Cycling stability performance and coulombic efficiency of PANI/g-POCN after 500 cycles. (**d**) Nyquist curves of Nanocomposites PANI/g-OCN and PANI/g-POCN with a fitted electrical equivalent circuit.

**Table 1 polymers-14-03913-t001:** Values of *C_s_*, Energy density, Power density, and electrical conductivity of Free PANI, PANI/g-OCN, and PANI/g-POCN.

Material	Conductivity (S/cm)	*C_s_* (F/g) 0.2 A/g	*C_s_* (F/g) 5 A/g
Free PANI	0.0045	322	82
PANI/g-OCN	0.0171	361	165
PANI/g-POCN	0.0375	450.5	294

## Data Availability

Data presented in this study are available on request from the corresponding author.

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
