# Peer review of "Biomimetic Synthesis of PANI/Graphitic Oxidized Carbon Nitride for Supercapacitor Applications"

_polymers, 2022, doi:10.3390/polym14183913_

Round 1
Reviewer 1 Report
Author reported “PANI-Graphitic oxidized carbon nitride composites as potential supercapacitor material”. Herein, the Reviewer feels that few corrections are necessary to improve the scientific quality of this paper, as suggested below. Questions and comments:
1. It is recommended to include a schematic diagram to depict the significant components of the system. Additionally, each figure needs to be polished.
2. Authors need to rewrite introduction part briefly and incorporate the following recent articles. Journal of materials chemistry A, 7(3), 946-957., Nano Energy, 77, 105276.
3. All Nyquist diagrams must be accompanied by analogue circuit diagrams according to Macromolecular Rapid Communications 43, no. 7 (2022): 2100905.
4. In the Electrode preparation and electrochemical measurements section, what about reference and counter electrode?
5. What about the coulomb efficiency?
6. Page 14, line 11, “conductivity of the polymers (see Table 7) and their adequate nanostructure” where is table 7?
7. Table showing comparison between current work and earlier relevant studies should be included.
8. There are more typing mistakes throughout the manuscript; the authors should carefully revise it. Like page 6, line 7 “For the calculation of the potential density”. What is potential density?
9. Authors are encouraged to cite some relevant and recent literature, such as.: Energy Storage Materials 42 (2021): 252-258. Materials Chemistry Frontiers, 5(3), 1324-1329. Materials science and engineering: B 262 (2020): 114766. And compare their results with other reported works.
10. Rate capability test must be included.
Overall suggestions and opinion:
The authors should thoroughly revise the manuscript keeping in view the aforementioned suggestions. The language should also need to be improved on many places.
Reviewer 2 Report
Cartagena et al. synthesized oxygen and phosphorous-modified carbon nitrides, which were used to biomimetic-assist aniline polymerization. The prepared PANI/g-OCN and PANI/g-POCN are used for supercapacitor applications. The synthesis of polymer composite synthesis and characterization are discussed fairly. Application part many weakly discussed. Therefore, I recommend a major revision for this work.
1. Page 11, line 393 author claimed “quasi-rectangular” for all three Free PANI, PANI/g-OCN and PANI/g-POCN modified electrodes. Indifferently, CV trace shows PANI/g-POCN only quasi-rectangular shape other two electrodes such as Free PANI, PANI/g-OCN seems like quasi reversible system. This should be corrected.
2. Page 12 author should provide the redox reaction mechanism as a chemical reaction for the reader’s best understanding in the page 12 discussion section.
3. Page 13, check current density value and unit, and page 14 line 496 “Table 7” should be “Table 1”.
4. The energy and power density calculations should be in a realistic two-electrode system, so I recommend removing energy and power density incorporation and discussion in the manuscript.
Reviewer 3 Report
The paper represents PANI-Graphitic oxidized carbon nitride composites as potential supercapacitor material. The biomimetic polymerization of aniline assisted by hematin, supported on phosphorous and oxygen modified carbon nitrides (g-POCN and g-OCN, respectively), achieved up to 89% yield. The obtained PAI/g-POCN and PANI/g-OCN show enhanced electrochemical properties, such as conductivity of up to 0.0375 S/cm, specific capacitances (Cs) of up to 294 F/g (at high current densities, 5 A/g) and a stable operation after 500 charge-discharge cycles (at 3 A/g). In contrast, the biomimetic synthesis of Free PANI, assisted by stabilized hematin in cosolvents, exhibited lower performance properties (65%). Due to their structural differences, the electrochemical properties of Free PANI (conductivity of 0.0045 S/cm and Cs of up to 82 F/g at 5 A/g) were lower than those of nanostructured PANI/g-POCN and g-OCN supports provide stability and improve the properties of biomimetically synthesized PANI. Overall the paper is very well organized and written. As such I did not find any short coming therefore it is accepted for publication. Just one small correction is needed in Figure 1 a . Please mention that the graph is UV-Vis spectrum.
Round 2
Reviewer 1 Report
Can be accept